# Translation, cultural adaptation, and validation of the Geriatric Depression Scale – Short Form into Bengali among elderly people in Bangladesh

Mahmud Hassan Arif[1]*, Panchanan Acharjee[1], Mohammad Waliul Hasnat Sajib[2], Md. Maruful Haque[3], Fahmida Hoque Rimti[1], Mohiuddin Ahsanul Kabir Chowdhury[4]

1 Chittagong Medical College, Chittagong, Bangladesh, 2 Tangail Medical College, Tangail, Bangladesh, 3 Shaheed Ziaur Rahman Medical College, Bogura, Bangladesh, 4 Asian University for Women, Chittagong, Bangladesh

* arif.mahmud@gmail.com

## Abstract

Depression is a major health concern among the elderly, affecting their quality of life. The Geriatric Depression Scale – Short Form (GDS-15) is commonly used to screen for depression in older adults. This study aimed to translate, culturally adapt, and validate the GDS-15 into Bengali for use among elderly people in Bangladesh. The GDS-15 was translated into Bengali, followed by cultural adaptation. A cross-sectional study was conducted with 100 elderly participants aged 60 years, and above in Chittagong Medical College Hospital. Psychometric validation was performed using reliability testing, chi-square analysis, ANOVA, and principal component analysis (PCA). The Bengali version of the GDS-15 demonstrated strong construct validity, with significant correlations between the original and translated versions of the items ($r = 0.9788$, $p < 0.001$). However, reliability analysis showed a low Cronbach's alpha (0.4753), indicating suboptimal internal consistency. The PCA revealed multiple components, suggesting the multidimensionality of the scale in this population. The Bengali GDS-15 showed low internal consistency which warrants the need for further cultural refinement and psychometric validation before routine use among the population. However, the Bengali GDS-15 has the potential to serve as a useful screening tool for geriatric depression in Bangladesh with further improvements.

## Introduction

Depression is a common mental health condition among elderly individuals, significantly impacting their quality of life, physical health, and overall well-being [1]. The prevalence of geriatric depression is particularly concerning in low- and middle-income countries, where the elderly population often faces unique challenges, such as limited access to healthcare, social isolation, and increased economic dependency [2]. In Bangladesh, these challenges are further exacerbated by cultural stigmatization of mental

**Data availability statement:** All relevant data are within the paper and its Supporting information files.

**Funding:** The authors received no specific funding for this work.

**Competing interests:** The authors have declared that no competing interests exist.

health issues, leading to underdiagnosis and undertreatment of depression among older adults [3]. Early detection of depressive symptoms is therefore crucial for improving the mental health and quality of life of elderly people [4].

The Geriatric Depression Scale (GDS) is one of the most widely used screening tools for detecting depression in elderly populations [5,6]. The Short Form of the GDS (GDS-15) is a concise version comprising 15 items, designed to facilitate rapid assessment in clinical and community settings [7,8]. Its brevity and ease of use make it particularly suitable for populations with low literacy and limited access to healthcare professionals, such as those in rural Bangladesh [9]. However, before the GDS-15 can be used effectively in a new cultural setting, it is essential to ensure that the tool is linguistically and culturally appropriate for the target population [10].

Translation and cultural adaptation are necessary to maintain the original tool's conceptual and functional equivalence in a different language and cultural context [11]. These processes help to ensure that the translated items accurately reflect the intended meanings and that the tool is appropriate for the cultural norms and experiences of the target population [11]. In addition, psychometric validation is needed to assess the reliability and validity of the translated version, ensuring that it measures depression accurately and consistently among the target population [12]. Adapting the Geriatric Depression Scale (both the 30- and 15-item versions) for use among older adults in Bangladesh requires rigorous linguistic and cultural validation. A Bangla translation already exists, including piloting to ensure technical and conceptual equivalence in local contexts [13]. More recently, psychometric evaluation of the Bangla GDS-SF (15-item) found strong internal consistency (Cronbach's $\alpha = 0.836$, McDonald's $\Omega = 0.841$), plus solid construct validity in both community and slum populations [9]. That study also identified key associated factors including age, gender (lower risk among men), and satisfaction with children's support, which align with international findings. Current practice in Bangladesh lacks a standardized geriatric depression screening tool, assessments are typically informal, verbal, and symptom-based, particularly among frontline health workers and NGOs. Rolling out the updated GDS would thus involve not only translation but also healthcare provider training, pilot testing in diverse settings, establishing cut-off thresholds aligned with local prevalence, and sensitization campaigns to reduce stigma and improve health literacy. Embedding it into routine geriatric assessments could significantly enhance early detection and referral pathways in both governmental and community-based care.

This study aimed to validate the GDS-15 short form for use among elderly individuals in Bangladesh which seeks to provide a valid and reliable screening tool for geriatric depression in the country, contributing to improved mental health care for the country's aging population.

## Materials and methods

### Study design and study setting

We conducted a cross-sectional study involving 98 participants aged 60 years or above who were admitted to the geriatric unit of the medicine ward in the Chittagong Medical College Hospital.

## Study population

The Inclusion criteria included the age ≥ 60 and the participants' ability to provide consent. Conversely, the exclusion criteria constituted severe cognitive impairment or acute psychiatric illness.

## Sample size and sampling technique

The study included a total of 98 elderly participants selected through purposive sampling to explore the socio-demographic, clinical, and lifestyle factors associated with depression in this population. The sample size was calculated considering the prevalence of depression among the elderly of 55.5% with a 95% confidence interval and a 10% margin of error. Participants were aged 60 years or older and were recruited from the CMCH, a tertiary health care centre, where people come from a diverse community to seek care. The study population included individuals with varying levels of education, marital status, family types, and comorbid conditions.

## Data collection instrument and cultural adaptation

The GDS-15 scale was utilized for data collection which was translated into Bengali using a standard process that involved forward translation, backward translation, and review by an expert committee comprising bilingual professionals. Cultural adaptation was conducted to ensure the scale was relevant to the context of Bangladeshi elderly individuals, including modifications to language and cultural expressions. To ensure cultural relevance and contextual appropriateness, the study followed a systematic cultural adaptation process alongside translation (Forward translation, reconciliation, and back translation). Initially, the original instrument and translated document were reviewed by a panel of bilingual experts in geriatric mental health and public health familiar with the local cultural context. This panel identified and discussed items that could be culturally sensitive or potentially misunderstood. Modifications were made to align concepts with local expressions, idioms, and social norms while retaining the original meaning. A pilot test was then conducted with a small group of older adults [5] representative of the target population to assess clarity, comprehension, and cultural resonance. Feedback from the pilot was incorporated to refine the instrument further. This iterative approach ensured that the adapted tool was not only linguistically accurate but also culturally appropriate for use in the Bangladeshi context.

## Data collection

The translated version of the GDS-15 was administered by trained interviewers. In addition to the GDS-15, socio-demographic data including age, sex, education level, and other relevant characteristics were collected. Before conducting the study, we conducted a pre-test among five (5) elderly patients attending the private chambers to check for cultural sensitivity.

## Statistical analysis

Data was analyzed using Stata BE18.0. [14]. Descriptive statistics were calculated for the socio- demographic characteristics of the participants. The reliability of the scale was assessed using Cronbach's alpha. Chi-square tests, correlation analyses, ANOVA, and principal component analysis (PCA) were conducted to validate the structure and consistency of the Bengali version of the GDS-15. We conducted a Pearson's Correlation test to identify the correlation between different components. Principal Component Analysis (PCA) was conducted to assess the underlying structure and internal consistency of the Bengali version of the Geriatric Depression Scale-15 (GDS-15). This statistical technique helped identify the latent components that contribute to the variance in item responses, thereby validating the dimensional structure of the scale in the local context. By examining factor loadings and eigenvalues, the PCA provided insight into how well the items grouped together to measure depressive symptoms among Bengali-speaking elderly individuals. The analysis also served to evaluate whether the translated version maintained conceptual alignment with the original instrument and whether the

items consistently captured the intended psychological constructs. We also measured the sensitivity and specificity of the translated tool in comparison with the non-translated, validated GDS-15 tool.

### Ethical approval

The ethical approval was obtained from the Institutional Review Board of the Chittagong Medical College Hospital. (Memo no: CMC/PG/2022/248).

## Results

### Socio-demographic characteristics

The study evaluated socio-demographic characteristics across three depression categories: no depression, mild depression, and moderate depression. Among ninety-eight study participants, 61.2% were diagnosed with mild depression and 34.7% were diagnosed with moderate depression. The mean age was 66.6 ± 4.4 years, with no statistically significant differences among the depression categories (p = 0.439). The proportion of males (77.5%) was higher than females (22.5%) across all groups, and sex differences were not statistically significant (p = 0.462). Most participants had completed 11–12 years or ≥13 years of education, and there was no significant association with depression severity (p = 0.667). Most participants lived in joint families (80.6%) and were married (77.6%). Religion, smoking status, and psychiatric or family history of psychiatric disorders showed no significant association with depression severity. The socio-demographic characteristics are detailed in Table 1.

### Lifestyle and health-related characteristics

There were no significant associations between smoking status and depression severity (p = 0.673). The presence of comorbid conditions was prevalent across all groups, with 96.9% of the total sample reporting at least one comorbidity (p = 0.375). Additionally, a family history of psychiatric disorders was uncommon, with only 6.1% of participants reporting such a history and no significant differences observed between depression categories (p = 0.863). Past psychiatric history was similarly low, with 5.1% of participants reporting a prior diagnosis, and this was not significantly associated with depression severity (p = 0.655). These findings suggest that while comorbid conditions are highly prevalent among the elderly, other factors may play a more significant role in the development and severity of depressive symptoms. The details are provided in Table 2.

### Internal consistency

The reliability analysis using Cronbach's alpha for the 15-item GDS-15 scale showed a coefficient of 0.5638, which indicates low internal consistency. The Bengali translated GDS- 15 scale yielded a Cronbach's alpha of 0.5408.

### Bivariate analysis

The bivariate analysis compared the agreement between the English and Bengali versions of GDS-15. High agreement was observed for all items, with percentages ranging from 84.6% to 98.4% (p < 0.001 for all) (Table 3).

### Diagnostic performance

Table 4 shows the diagnostic accuracy of the translated scale. The GDS-15 scale demonstrated good diagnostic accuracy. Sensitivity for detecting mild depression was 100%, and specificity was 75%, with an ROC area of 87.5% (63.0–100.0%) (Fig 1). For moderate/severe depression, sensitivity was 94.1%, specificity was 84.4%, and the ROC area was 89.3% (83.3–95.3%) (Fig 2).

**Table 1. Socio-demographic characteristics of the participants (N = 98).**

| Characteristics | No Depression % (n) | Mild Depression % (n) | Moderate Depression % (n) | Total % (n) | P-Value |
|---|---|---|---|---|---|
| Age, in years (Mean ± SD) | 64.3 ± 3.0 | 67.2 ± 4.2 | 65.8 ± 4.8 | 66.6 ± 4.4 | 0.439[†] |
| 60–64 years | 25.0 (1) | 33.0 (20) | 50.0 (17) | 38.8 (38) | |
| 65–69 years | 75.0 (3) | 33.3 (20) | 29.4 (10) | 33.7 (33) | 0.360[‡] |
| 70–74 years | 0.0 (0) | 26.7 (16) | 14.7 (5) | 21.4 (21.0) | |
| More than 75 years | 0.0 (0) | 6.7 (4) | 5.9 (2) | 6.1 (6) | |
| Sex | | | | | |
| Male | 75.0 (3) | 81.7 (49) | 31.4 (24) | 77.5 (76) | 0.462[‡] |
| Female | 25.0 (1) | 18.3 (11) | 29.4 (10) | 22.5 (22) | |
| Education | | | | | |
| 6–10 years | 0.0 (0) | 5.0 (3) | 2.9 (1) | 4.1 (4) | 0.667[‡] |
| 11–12 years | 25.0 (1) | 45.0 (27) | 55.9 (19) | 47.9 (47) | |
| ≥ 13 years | 75.0 (3) | 50 (30.0) | 41.2 (14) | 47.9 (47) | |
| Family type | | | | | |
| Nuclear | 25.0 (1) | 16.7 (10) | 11.8 (4) | 15.3 (15) | 0.672[‡] |
| Joint | 75.0 (3) | 76.7 (46) | 88.3 (30) | 80.6 (79) | |
| Extended | 0.0 (0) | 6.7 (4) | 0.0 (0) | 4.1 (4) | |
| Marital status | | | | | 0.708[‡] |
| Married | 3 (75.0) | 45 (75.0) | 28 (82.4) | 76 (77.6) | |
| Widowed/Widower | 1 (25.0) | 15 (25.0) | 6 (17.7) | 22 (22.5) | |
| Religion | | | | | |
| Muslim | 3 (75.0) | 48 (80.0) | 30 (88.2) | 81 (82.7) | 0.802[‡] |
| Non-Muslim | 1 (25.0) | 12 (20.0) | 4 (11.8) | 17 (17.3) | |

[†] t-test.

[‡] Chi-squared test.

**Table 2. Lifestyle and health-related characteristics (N = 98).**

| Characteristics | No Depression % (n) | Mild Depression % (n) - | Moderate Depression % (n) | Total % (n) | P-Value |
|---|---|---|---|---|---|
| Smoking status | | | | | |
| Non-smoker | 50.0 (2) | 38.3 (23) | 47.1 (16) | 41.8 (41) | 0.673[‡] |
| Ex- or Current Smoker | 50.0 (2) | 61.7 (37) | 52.9 (18) | 58.2 (57) | |
| Co-morbid condition | | | | | |
| No | 0.0 (0) | 5.0 (3) | 0.0 (0) | 3.1 (3) | 0.375[‡] |
| Yes | 100.0 (4) | 95.0 (57) | 100.0 (34) | 96.9 (95) | |
| Family History of Psychiatric Disorder | | | | | |
| No | 100.0 (4) | 93.3 (56) | 94.1 (32) | 93.8 (92) | 0.863[†] |
| Yes | 0.0 (0) | 6.7 (4) | 5.9 (2) | 6.1 (6) | |
| Past History of Psychiatric Disorder | | | | | |
| No | 100.0 (4) | 93.3 (56) | 97.1 (33) | 93.8 (92) | 0.655[†] |
| Yes | 0.0 (0) | 6.7 (4) | 2.9 (1) | 6.1 (6) | |

[‡] Chi-squared test [†] Fishers exact test.

**Table 3. Comparison between English and Bengali version of GDS scale [Presented as % (n)].**

| Questions | GDS English % (n) | GDS Translated % (n) | Agreement | P-value |
|---|---|---|---|---|
| Are you basically satisfied with your life? | 61.2 (60) | 62.6 (61) | 98.4% | <0.001 |
| Have you dropped many of your activities and interests? | 76.5 (75) | 76.5 (75) | 97.3% | <0.001 |
| Do you feel that your life is empty? | 32.7 (32) | 33.7 (33) | 97.0% | <0.001 |
| Do you often get bored? | 41.4 (41) | 42.9 (42) | 97.6% | <0.001 |
| Are you in good spirits most of the time? | 45.9 (45) | 46.9 (46) | 97.8% | <0.001 |
| Are you afraid that something bad is going to happen to you? | 58.2 (57) | 57 (56) | 98.2% | <0.001 |
| Do you feel happy most of the time? | 56.1 (55) | 61.2 (60) | 90.0% | <0.001 |
| Do you often feel helpless? | 60.2 (59) | 59.2 (58) | 98.3% | <0.001 |
| Do you prefer to stay at home, rather than going out and doing new things? | 66.3 (65) | 69.4 (68) | 94.1% | <0.001 |
| Do you feel you have more problems with memory than most people? | 54.1 (53) | 57.1 (56) | 87.5% | <0.001 |
| Do you think it is wonderful to be alive? | 56.1 (55) | 61.2 (60) | 88.3% | <0.001 |
| Do you feel pretty worthless the way you are now? | 46.9 (46) | 53.1 (52) | 84.6% | <0.001 |
| Do you feel full of energy? | 37.8 (37) | 39.8 (39) | 92.3% | <0.001 |
| Do you feel that your situation is hopeless? | 53.1 (52) | 55.1 (54) | 92.3% | <0.001 |
| Do you think that most people are better off than you are? | 42.9 (42) | 46.9 (46) | 89.1% | <0.001 |

**Table 4. Sensitivity and specificity of translated GDS scale.**

| Diagnosis | Sensitivity | Specificity | PPV | NPV | ROC area |
|---|---|---|---|---|---|
| Mild Depression | 100% | 75% | 99.0% | 100% | 87.5 (63.0–100.0)% |
| Moderate/Severe Depression | 94.1% | 84.4% | 76.2% | 96.4% | 89.3 (83.3–95.3)% |

## Factor analysis

Principal component analysis (PCA) revealed that five components had eigenvalues greater than 1, explaining 52.47% of the cumulative variance. The first component explained 15.3%, suggesting that the scale captures multiple dimensions of depression. Variance proportions were similar between the English and Bengali versions, indicating comparable dimensionality (Table 5).

## Cultural adaptation

Cultural adaptation ensured that the Bengali-translated GDS-15 reflected local expressions and culturally relevant concepts, enhancing its applicability to the elderly population in Bangladesh. An expert committee reviewed and supported the comprehensibility and cultural appropriateness of the translated items.

## Discussion

The present study aimed to evaluate the reliability, validity, and cultural adaptation of the Geriatric Depression Scale-15 (GDS-15) for use among elderly individuals in Bangladesh. The validation of the Bengali version of the GDS-15 in this study contributes to the growing body of evidence on adapting geriatric depression screening tools for culturally diverse populations. While the tool showed high diagnostic accuracy and strong agreement between the original and translated versions, the low internal consistency suggests challenges common to similar cross-cultural adaptations. For instance, studies in non-Western settings, such as the Igbo adaptation by Mgbeojedo et al. (2022) and Sultana et al. (2022) in

**Global Public Health**
PLOS

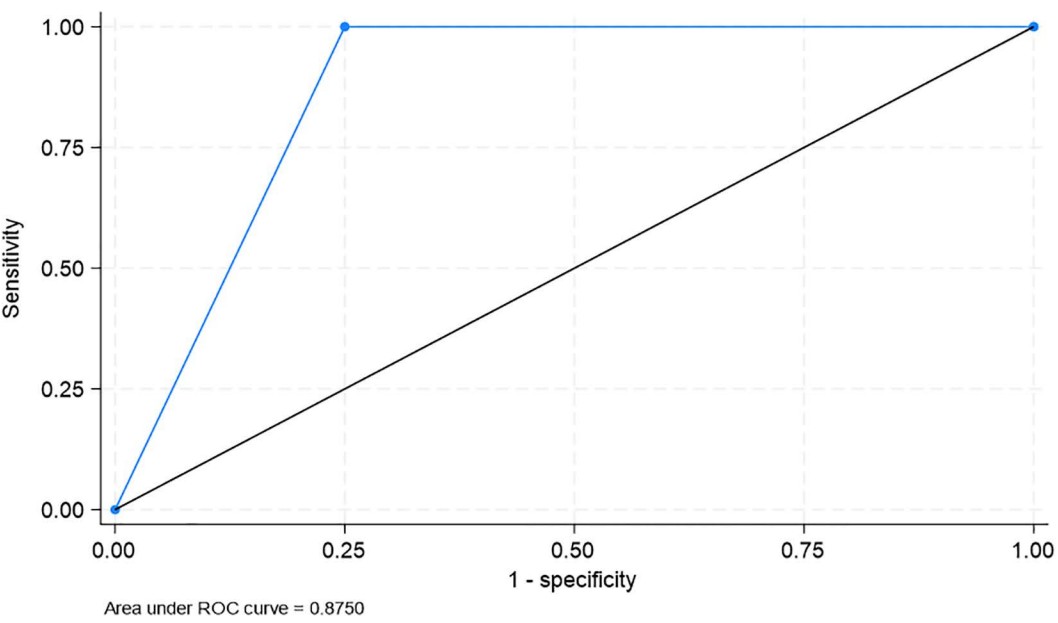

Area under ROC curve = 0.8750

**Fig 1. ROC Curve for mild depression.**

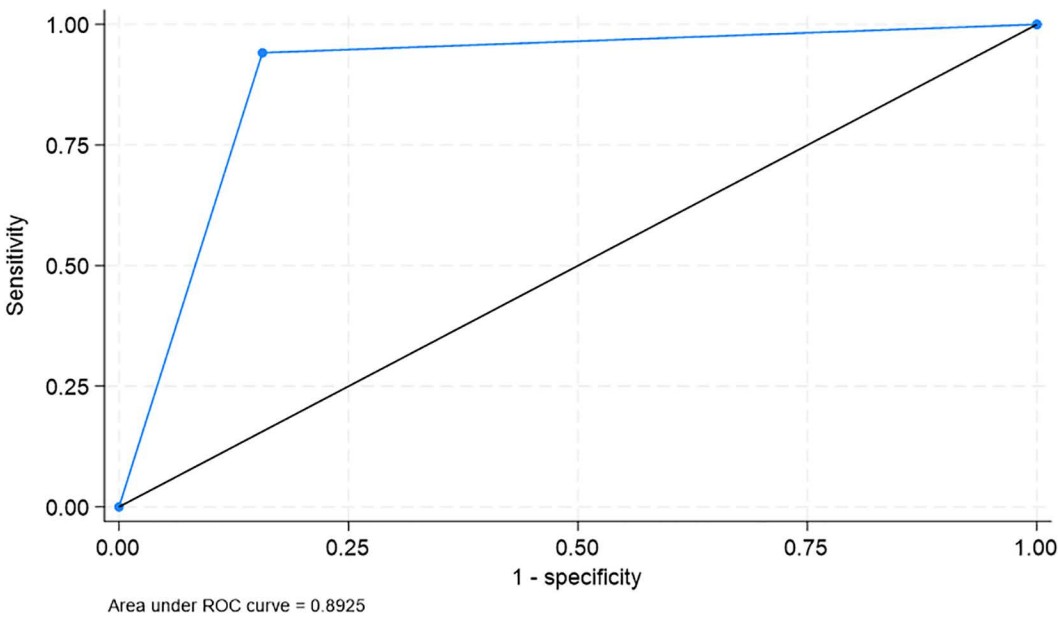

Area under ROC curve = 0.8925

**Fig 2. ROC Curve for moderate/severe depression.**

Bangladesh, similarly reported varying levels of reliability due to cultural nuances and differing interpretations of specific items [9,10].

The socio-demographic analysis showed no significant associations between depression severity and variables such as age, sex, education, marital status, or family structure. This suggests that depression among the elderly in

**Table 5. Principal component analysis of the GDS scale.**

| Questions | GDS English | | GDS Translated | |
|---|---|---|---|---|
| | Eigenvalue | Proportion (%) | Eigenvalue | Proportion (%) |
| Are you basically satisfied with your life? | 2.52 | 16.8 | 2.45 | 16.3 |
| Have you dropped many of your activities and interests? | 1.59 | 10.6 | 1.71 | 11.4 |
| Do you feel that your life is empty? | 1.50 | 10.0 | 1.47 | 9.8 |
| Do you often get bored? | 1.37 | 9.2 | 1.43 | 9.6 |
| Are you in good spirits most of the time? | 1.18 | 7.9 | 1.05 | 7.0 |
| Are you afraid that something bad is going to happen to you? | 1.05 | 7.0 | 0.99 | 6.6 |
| Do you feel happy most of the time? | 0.95 | 6.4 | 0.95 | 6.3 |
| Do you often feel helpless? | 0.89 | 5.9 | 0.89 | 6.0 |
| Do you prefer to stay at home, rather than going out and doing new things? | 0.78 | 5.2 | 0.82 | 5.5 |
| Do you feel you have more problems with memory than most people? | 0.69 | 4.6 | 0.72 | 4.8 |
| Do you think it is wonderful to be alive? | 0.62 | 4.1 | 0.67 | 4.5 |
| Do you feel pretty worthless the way you are now? | 0.59 | 3.9 | 0.59 | 3.9 |
| Do you feel full of energy? | 0.49 | 3.3 | 0.49 | 3.3 |
| Do you feel that your situation is hopeless? | 0.41 | 2.7 | 0.44 | 2.9 |
| Do you think that most people are better off than you are? | 0.37 | 2.4 | 0.33 | 2.2 |

this population is likely influenced by other factors, such as social isolation, chronic illness, or cultural perceptions of mental health, which were not directly assessed in this study [15–17]. However, there are a few studies that documented the association of geriatric depression with socio-demographic factors [17–21]. For instance, a survey conducted among the Thai elderly population found an association of depression with old age, being single, drinking alcohol daily, and having diabetes [18]. Another study showed that marital status is somehow linked to geriatric depression [19]. In Bangladesh, a study documented similar results where being without a spouse was identified as a risk factor [17]. A study looked into the risk factors for depression among the elderly population in Bangladesh and found similar results to this study [21].

The low Cronbach's alpha values (0.5638 for the English version and 0.5408 for the Bengali version) highlight the need for scale refinement, a limitation noted in comparable research. While these values fall below the commonly accepted threshold of 0.7, they are equivalent to other studies evaluating the GDS in culturally diverse settings. This result may indicate that some items on the scale are less relevant or less consistently understood in the Bangladeshi context. For example, questions reflecting specific cultural or social norms might contribute to lower reliability. In Krishnamoorthy et al.'s meta-analysis (2020), the reliability of the GDS-15 varied across cultures, with adaptations often requiring modifications to address local norms [6]. Refinement of the scale, such as revising or removing items with low item- total correlations, may improve its psychometric properties.

Despite low internal consistency, the translated GDS-15 demonstrated strong agreement with the original English version, with all items showing high concordance (≥84.6%). This suggests that the translation process successfully preserved the content and intent of the original scale. Furthermore, the tool demonstrated excellent diagnostic performance, with high sensitivity and specificity for detecting mild and moderate/severe depression. The ROC areas (87.5% for mild and 89.3% for moderate/severe depression) underscore the effectiveness of the GDS-15 as a screening tool in this population. The high sensitivity and specificity of the Bengali GDS-15 (87.5% for mild depression and 89.3% for moderate/severe depression) are consistent with other regional adaptations, such as those observed in Sultana et al.'s (insert year) study in Bangladesh [9]. This underscores the utility of the GDS-15 as a screening tool in low-resource settings where mental health services are limited. However, the identified multidimensionality suggests that a more tailored approach may be

needed to fully capture the construct of depression in the Bangladeshi elderly population. These findings are consistent with previous studies that validated the GDS-15 in other non-Western settings.

The factor analysis depicted that depression in this population is multidimensional, with five components explaining 52.47% of the total variance. The first component accounted for only 15.3% of the variance, suggesting that the scale does not capture a single dominant factor of depression. This is an important finding, as it indicates that the construct of depression among elderly individuals in Bangladesh may differ from that in Western populations, where the GDS was originally developed. The identified dimensions may reflect unique cultural and social determinants of mental health in this setting, warranting further qualitative research.

This finding emphasizes the multidimensional nature of depression in non-Western contexts, as indicated by the principal component analysis in this study, which identified five distinct factors accounting for 52.47% of the variance. Such multidimensionality aligns with findings from Shin et al. (2019), who observed variations in depressive symptomatology influenced by social and cultural factors [8].

The cultural adaptation process was instrumental in ensuring the scale's comprehensibility and relevance to the elderly Bangladeshi population. Modifications to language and phrasing were guided by an expert committee, incorporating cultural nuances and local expressions. This adaptation likely contributed to the high agreement between the original and translated versions. However, the lower internal consistency highlights the need for further adjustments to improve the scale's reliability without compromising its cultural appropriateness.

### Strengths and limitations

A key strength of this study was the rigorous translation and cultural adaptation process, which ensured that the Bengali GDS-15 reflected local linguistic and cultural expressions. Similar methodologies have been advocated by Borsa et al. (2012) and Arafat et al. (2016), who stressed the importance of preserving conceptual equivalence in cross-cultural validations [11,12]. Additionally, the inclusion of a diverse elderly population enhances the generalizability of the findings within Bangladesh.

Despite these efforts, some items may have been less culturally relevant, potentially contributing to lower internal consistency. Future revisions should consider qualitative methods, such as focus groups, to explore cultural perceptions of the scale's items. Several other limitations should be noted. First, the sample size, while adequate for the analyses conducted, may not capture the full heterogeneity of the elderly population. Second, the study relied on self-reported data, which may be subject to social desirability bias, particularly in a cultural context where mental health remains stigmatized. Finally, the low internal consistency of the GDS-15 indicates that additional refinement of the scale is necessary for optimal use in this setting.

### Future directions

Future research should explore qualitative approaches to better understand the dimensions of depression among elderly individuals in Bangladesh. Focus group discussions or in-depth interviews could help identify culturally specific factors influencing depressive symptoms, which may inform the development of a more tailored assessment tool. Additionally, longitudinal studies are needed to examine the predictive validity of the GDS-15 and its utility in tracking changes in depressive symptoms over time.

### Conclusion

In conclusion, the GDS-15 demonstrates promise as a screening tool for depression among elderly individuals in Bangladesh. While the translation and cultural adaptation process ensured high agreement and diagnostic performance, the low internal consistency underscores the need for further refinement of the scale. Addressing these limitations will enhance the utility of the GDS-15 in identifying and addressing mental health needs in this vulnerable population, contributing to improved geriatric care and mental health outcomes in Bangladesh.

## Supporting information

**S1 Data. The dataset for this study.**
(XLSX)

## Acknowledgments

We want to acknowledge the authority of Chittagong Medical College Hospital to allow us to conduct the study in the hospital. We are also grateful to the caregivers of the elderly patients admitted to the hospital. Furthermore, the contribution of the geriatric population in this paper is invaluable.

## Author contributions

**Conceptualization:** Mahmud Hassan Arif.

**Data curation:** Mahmud Hassan Arif, Fahmida Hoque Rimti.

**Formal analysis:** Mohiuddin Ahsanul Kabir Chowdhury.

**Investigation:** Panchanan Acharjee, Mohammad Waliul Hasnat Sajib, Md. Maruful Haque, Fahmida Hoque Rimti.

**Methodology:** Mahmud Hassan Arif, Panchanan Acharjee, Mohammad Waliul Hasnat Sajib, Md. Maruful Haque.

**Project administration:** Mahmud Hassan Arif.

**Software:** Mohiuddin Ahsanul Kabir Chowdhury.

**Supervision:** Mahmud Hassan Arif, Panchanan Acharjee.

**Validation:** Panchanan Acharjee, Mohiuddin Ahsanul Kabir Chowdhury.

**Writing – original draft:** Mohiuddin Ahsanul Kabir Chowdhury.

**Writing – review & editing:** Mahmud Hassan Arif, Panchanan Acharjee.

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
