## [Decision Letter · Decision Letter 0]

5 Jun 2025

PGPH-D-25-00588

Translation, Cultural Adaptation, and Validation of the Geriatric Depression Scale – Short Form into Bengali among Bangladeshi Elderly People

Dear Dr. Arif,

Thank you for submitting your manuscript to PLOS Global Public Health. After careful consideration, we feel that it has merit but does not fully meet PLOS Global Public Health’s publication criteria as it currently stands. Therefore, we invite you to submit a revised version of the manuscript that addresses the points raised during the review process.

EDITOR:

Authors should pay particular attention to the comments regarding methods. The translation procedure should be detailed and the guidelines followed to complete it should be mentioned. Authors should consider using EFA/CFA to explore factor structure, rather than PCA.

We look forward to receiving your revised manuscript.

Kind regards,

Feten Fekih-Romdhane

Academic Editor

Journal Requirements:

1. In the online submission form, you indicated that “available on request”.

a. In a public repository,

b. Within the manuscript itself, or

c. Uploaded as supplementary information.

Additional Editor Comments (if provided):

Reviewers' comments:

Reviewer's Responses to Questions

**Comments to the Author**

1. Does this manuscript meet PLOS Global Public Health’s publication criteria?

Reviewer #1: Partly

Reviewer #2: Yes

Reviewer #3: Yes

Reviewer #4: No

Reviewer #5: No

2. Has the statistical analysis been performed appropriately and rigorously?

Reviewer #1: Yes

Reviewer #2: Yes

Reviewer #3: Yes

Reviewer #4: Yes

Reviewer #5: I don't know

3. Have the authors made all data underlying the findings in their manuscript fully available (please refer to the Data Availability Statement at the start of the manuscript PDF file)?

Reviewer #1: Yes

Reviewer #2: No

Reviewer #3: Yes

Reviewer #4: No

Reviewer #5: No

4. Is the manuscript presented in an intelligible fashion and written in standard English?

Reviewer #1: Yes

Reviewer #2: Yes

Reviewer #3: Yes

Reviewer #4: Yes

Reviewer #5: Yes

Reviewer #1: This study aimed to translate, culturally adapt, and validate the GDS-15 into Bengali for use among elderly people in Bangladesh. It is suggested to revise the manuscript to better provides useful knowledge addition to this topic area.

①The material and translation sections are not described in sufficient detail. It is recommended to provide detailed steps for translating the GDS-15 questionnaire into Bengali. More specifics should be provided to ensure repeatability.

②Was there a pre-survey after the scale was translated? Is there a correlation between the low Cronbach's alpha and pre-survey and sample size? It was suggested that further elaboration be provided in the discussion section.

Reviewer #2: Overall, this article is well-written and clinically relevant as it offers a potentially useful screening tool for potentially underserved elderly populations in Bangladesh. However, several areas require attention to strengthen the manuscript. Specifically, the introduction needs better contextualization, the methods require clarification on sampling, sample size, and expert committee composition, and the results should include more detail on statistical approaches and limitations. Addressing these points would significantly enhance the paper’s rigor and impact.

Introduction

• Provide additional context on depression among elderly Bangladeshis (e.g., prevalence rates, risk factors) to underscore the study’s importance.

• Discuss prior Bengali translations of the GDS-SF (e.g., Lahiri & Chakraborty’s work in West Bengal) and how this study builds on or differs from existing literature.

Methods

• Fix typos (e.g., "60 years I the Chittagong Medical College Hospital" → "60 years in the Chittagong Medical College Hospital").

• Clarify composition of the expert committee (e.g., mental health professionals, linguists) and steps taken to mitigate bias (e.g., interrater reliability checks). Was feedback from Bangladeshi elders incorporated during cultural adaptation?

• Explain purposeful sampling strategy (e.g., quotas, inclusion/exclusion criteria).

• The title states "hospital-based study," but the methods mention community recruitment. Resolve this discrepancy.

• If the study was conducted in a hospital setting, discuss potential selection bias (e.g., mental health stigma deterring participation in a hospital-based study)?

• Was a power analysis conducted to justify the sample size (n = 98)?

Results

• Only 4 participants were in the "no depression" group (Table 1). Discuss implications for generalizability.

• Include race/ethnicity breakdown if collected.

• Bivariate analysis: Clarify whether participants completed both English and Bengali GDS versions (unclear in current text).

• PCA:

o Specify the 5 components and their alignment with the original English scale.

o Discuss cultural differences in item interpretation.

o Justify PCA over EFA/CFA, as the latter are more common for validation studies.

Discussion

• Given low internal consistency, report item-total correlations to identify weak items (e.g., "Which items contributed to low α?"). Propose concrete revisions to improve internal consistency.

• Expand on sampling bias (hospital vs. community) and low "no depression" group representation.

Overall, the study is promising but requires methodological clarifications and deeper discussion of limitations to meet its full potential. With these revisions, it would be a stronger contribution to geriatric mental health research in Bangladesh.

Reviewer #3: General comments and some key concerns:

The manuscript is addressing an important issue on “Bangali translation of the short form Geriatric Depression Scale, Cultural Adaptation, and Validation among Bangladeshi Elderly People”. Geriatric Depression is a common problem globally and more so among Bangladeshi Elderly People. However, there are some comments that need to be addressed as below.

• The authors should ensure grammar issues are ironed out in the whole document i.e. the sentence following the Study Design and Study setting is not clear.

1. Title: The authors need to modify the title so that it is clear and the following is the suggestion ““Bengali translation of short form Geriatric Depression Scale, Cultural Adaptation, and Validation among Bangladeshi Elderly People: a hospital-based study”

2. Introduction

• The authors should provide more information on the 15 items of the Short Form of the GDS (GDS-15) in the introduction. They should also explain the knowledge gap on the use of the GDS-15 tool among Bangladeshi Elderly People since it is already validated tool in use to assess Geriatric Depression.

3. Methods section

• The authors should change materials and methods to “Methodology”. The sentence under study design and study setting is not clear and it needs to be paraphrased. The authors should elaborate more on the Chittagong Medical College Hospital i.e. where is it found, what is its bed capacity? They should also include the study population and why the study population was considered for this study. The authors should include the selection criteria of the study participants including both the inclusion and exclusion criteria. The sentence on data collection needs to be paraphrased since they are not clear. The authors should also explain how the data collected was managed and ensured quality control. The sub-sections in the methodology need to be re-arranged such as the sample size and sampling technique should come before data collection procedure.

4. Results

• Since the main measure of reporting the findings is percentage (%) and not frequencies, therefore the authors should report the findings as % (n) and not n(%) and this should be corrected throughout the document and most especially the tables in the result section. The sub-titles in the different column of the tables should be written to include % (n) for more clarity. Since years in the row of age in table 1 has already been included in the sub-title, then year should be removed from the different values. Then in the education row, what are the different values in years mean in table 1. In the family type, the authors should explain what they mean by “joint family” in relation to the classification or types of the families. The same above issues should also be applied to table 2. In the internal consistency, the reliability analysis using Cronbach's alpha for the 15-item GDS-15 scale revealed a coefficient of 0.5638, why was this low when the GDS-15 tool is already validated and used internationally as a tool for assessing geriatric depression. In table 4, the % should be removed from the values and put on the sub-titles.

5. Conclusion

• The statement in conclusion, “the GDS-15 demonstrates promise as a screening tool for depression among elderly individuals in Bangladesh”. Does this mean that the use of GDS-15 tool as standard tool in assessing geriatric depression is not standardized? And therefore should not be relied on to assess geriatric depression.

Reviewer #4: though they have done rigorous data analysis but still they need to further analysis the data and see if Exploratory Factor Analysis (EFA) could be done to enhance the validity of the tool.

Data is available on request as mentioned by the author in the Data Availability Statement at the start of the manuscript PDF file.

The reliability is low 0.475 and reported differently at various part of the manuscript. In conclusion, it's written that the low internal consistency underscores the need for further refinement of the scale.

Reviewer #5: It is unclear what methodologies were used for adaptation. Given the lack of clarity on methodology and what methods were used to reach which conclusions, it does not appear data fully support conclusions.

**Do you want your identity to be public for this peer review?** For information about this choice, including consent withdrawal, please see our Privacy Policy

Reviewer #1: No

Reviewer #2: No

Reviewer #3: No

Reviewer #4: No

Reviewer #5: No

---

## [Decision Letter · Decision Letter 1]

9 Sep 2025

PGPH-D-25-00588R1

Translation, Cultural Adaptation, and Validation of the Geriatric Depression Scale – Short Form into Bengali among Bangladeshi Elderly People

Dear Dr. Arif,

Thank you for submitting your manuscript to PLOS Global Public Health. After careful consideration, we feel that it has merit but does not fully meet PLOS Global Public Health’s publication criteria as it currently stands. Therefore, we invite you to submit a revised version of the manuscript that addresses the points raised during the review process.

The manuscript has been evaluated by three of the original reviewers, and their comments are available below.

Could you please revise the manuscript to carefully address the concerns raised?

We look forward to receiving your revised manuscript.

Kind regards,

Steve Zimmerman

Staff Editor

Journal Requirements:

Additional Editor Comments (if provided):

Reviewers' comments:

Reviewer's Responses to Questions

**Comments to the Author**

Reviewer #1: All comments have been addressed

Reviewer #3: All comments have been addressed

Reviewer #4: All comments have been addressed

publication criteria?

Reviewer #1: Yes

Reviewer #3: Partly

Reviewer #4: Yes

3. Has the statistical analysis been performed appropriately and rigorously?

Reviewer #1: Yes

Reviewer #3: Yes

Reviewer #4: No

4. Have the authors made all data underlying the findings in their manuscript fully available (please refer to the Data Availability Statement at the start of the manuscript PDF file)?

Reviewer #1: Yes

Reviewer #3: Yes

Reviewer #4: Yes

5. Is the manuscript presented in an intelligible fashion and written in standard English?

Reviewer #1: Yes

Reviewer #3: Yes

Reviewer #4: Yes

Reviewer #1: 1. In the "Materials and Methods" section, all the subjects were from geriatric wards. The sample was overly concentrated in a specific group. Was it representative? Were all the subjects suffering from chronic diseases? Chronic disease patients may be more prone to depressive symptoms due to physical discomfort and other factors. Would this affect the reliability and validity of the scale?

2. The subjects were excluded only if they had severe cognitive impairments or acute mental illnesses. All the subjects were from geriatric wards. Were they also suffering from severe physical diseases, chronic mental illnesses, language disorders, etc.? Would this affect the validity of the scale?

3. The pre-test only involved 5 elderly patients. Were the cultural adaptability issues fully exposed? Would this affect the reliability and validity of the scale?

Reviewer #3: General comments and some key concerns:

The paper is addressing an important topic on “Translation, Cultural Adaptation, and Validation of the Geriatric Depression Scale –Short Form into Bengali among Bangladeshi Elderly People”. However, there are some issues that need to be addressed as indicated.

1. General comments

It is an interesting study providing the status of mental illness among patients on opioid agonist treatment programmes in Ukraine during the conflict. However, there are some issues that need to be addressed as indicated.

• There are some grammar issues that need to be corrected in the paper.

• The authors should ensure uniformity in the referencing style throughout the whole paper.

2. Title: Need to be improved. Suggestion “Translation into Bengali language, Cultural Adaptation, and Validation of the short form Geriatric Depression Scale among Elderly People in Bangladeshi ”

3. Abstract

The authors need to explain what Bangali mean in the paper. The authors need to draw the conclusion based on the findings, which is not the case in the present paper

4. Introduction

The knowledge gap on the assessment of depression using Geriatric Depression Scale among the elderly people in Bangladesh is not clear in the introduction and therefore, the authors need to clearly explain the gap.

5. Materials and Methods

In this sub-section, the authors should use “Methodology” instead of the “Materials and Methods” because of the nature of the study. The first sentence on the Study Design and Study setting has grammar issues that the authors need to address. Further information on the study setting, study population, selection criteria and sampling procedure need to be elaborated in the paper. The authors need to explain how the GDS-15 was translated and by who?

6. Results

The table 1 needs to be revised to make it clear and the authors should explain what the values in the brackets mean and the units used. The authors should report the values as % (n) and not n (%) in all the tables (Table 1, 2, 3); if actually that is what they represent in the paper. Then the % on each value in table 3 and 4 should be removed and incorporated in the sub-title; similar to table 5. However, in the table 5, the values should be presented as % (n), and the authors have only presented % values.

7. Discussion

The authors need to have uniformity in the referencing style (see first paragraph on page 16).

8. Conclusion

The conclusion should be based on the findings from the study which is not the case in the present paper.

Reviewer #4: The comments given previously to the authors has been addressed. There is another important point to consider is that the content validation is missing. If the tool is translated from English into the local language, then Content Validity Index (CVI) needs to be consider to ensure its accuracy and appropriateness of the translation. The table of CVI from experts will be a good addition to the study.

**Do you want your identity to be public for this peer review?** For information about this choice, including consent withdrawal, please see our Privacy Policy

Reviewer #1: No

Reviewer #3: No

Reviewer #4: No

---

## [Decision Letter · Decision Letter 2]

18 Nov 2025

PGPH-D-25-00588R2

Translation, Cultural Adaptation, and Validation of the Geriatric Depression Scale – Short Form into Bengali among Bangladeshi Elderly People

Dear Dr. Arif,

Thank you for submitting your manuscript to PLOS Global Public Health. After careful consideration, we feel that it has merit but does not fully meet PLOS Global Public Health’s publication criteria as it currently stands. Therefore, we invite you to submit a revised version of the manuscript that addresses the points raised during the review process.

We look forward to receiving your revised manuscript.

Kind regards,

Theingi Maung Maung, Ph.D

Academic Editor

Journal Requirements:

Additional Editor Comments (if provided):

Reviewers' comments:

Reviewer's Responses to Questions

**Comments to the Author**

Reviewer #3: All comments have been addressed

Reviewer #4: All comments have been addressed

publication criteria?

Reviewer #3: Partly

Reviewer #4: Yes

3. Has the statistical analysis been performed appropriately and rigorously?

Reviewer #3: Yes

Reviewer #4: Yes

4. Have the authors made all data underlying the findings in their manuscript fully available (please refer to the Data Availability Statement at the start of the manuscript PDF file)?

Reviewer #3: Yes

Reviewer #4: Yes

5. Is the manuscript presented in an intelligible fashion and written in standard English?

Reviewer #3: Yes

Reviewer #4: Yes

Reviewer #3: The authors have tried to address the comments raised. However, the authors have not put attention to all the comments that were suggested i.e. Title: “Translation into Bengali language, Cultural Adaptation, and Validation of the short form Geriatric Depression Scale among Elderly People in Bangladeshi ” was suggested but was not effected.

Abstract

The authors need to draw the conclusion based on the findings, which is not the case in the present paper. This was not effected as well.

Results

The table 1 needs to be revised to make it clear and the authors should explain what the values in the brackets mean. Do the authors mean n(%) in table 1? Suggestion, the subheading in the first row for the different column should include % (n). Otherwise, these values do not make sense. In table 3 and 4, the authors have not responded to the following comments. The values should be presented as % (n).

Reviewer #4: The authors have completed some essential processes needed to validate their tool like PCA and reliability. There are a few steps that could have been included but the processes completed are enough to determine the validation.

**Do you want your identity to be public for this peer review?** For information about this choice, including consent withdrawal, please see our Privacy Policy

Reviewer #3: No

Reviewer #4: No

---

## [Editor Report · Decision Letter 3]

23 Dec 2025

PGPH-D-25-00588R3

Translation, Cultural Adaptation, and Validation of the Geriatric Depression Scale – Short Form into Bengali among Bangladeshi Elderly People

Dear Dr. Arif,

Thank you for submitting your manuscript to PLOS Global Public Health. After careful consideration, we feel that it has merit but does not fully meet PLOS Global Public Health’s publication criteria as it currently stands. Therefore, we invite you to submit a revised version of the manuscript that addresses the points raised during the review process.

Certain reviewer comments need to be addressed, including revisions to the abstract’s conclusion and modifications in the results section.

We look forward to receiving your revised manuscript.

Kind regards,

Theingi Maung Maung, Ph.D

Academic Editor
---

## [Editor Report · Decision Letter 4]

25 Jan 2026

PGPH-D-25-00588R4

Translation, Cultural Adaptation, and Validation of the Geriatric Depression Scale – Short Form into Bengali among Bangladeshi Elderly People

Dear Dr. Arif,

Thank you for submitting your manuscript to PLOS Global Public Health. After careful consideration, we feel that it has merit but does not fully meet PLOS Global Public Health’s publication criteria as it currently stands. Therefore, we invite you to submit a revised version of the manuscript that addresses the points raised during the review process.

Although the authors have addressed nearly all reviewer comments, minor revisions are still needed in the title, abstract, and results sections.

We look forward to receiving your revised manuscript.

Kind regards,

Theingi Maung Maung, Ph.D

Academic Editor

Journal Requirements:

Additional Editor Comments (if provided):

Although the authors have addressed nearly all reviewer comments, minor revisions are still needed in the title, abstract, and results sections.
---

## [Editor Report · Decision Letter 5]

23 Feb 2026

Translation, Cultural Adaptation, and Validation of the Geriatric Depression Scale – Short Form into Bengali among Elderly People in Bangladesh

PGPH-D-25-00588R5

Dear Dr. Arif,

We are pleased to inform you that your manuscript 'Translation, Cultural Adaptation, and Validation of the Geriatric Depression Scale – Short Form into Bengali among Elderly People in Bangladesh' has been provisionally accepted for publication in PLOS Global Public Health.

Best regards,

Theingi Maung Maung, Ph.D

Academic Editor
